# A Brief Review of Formaldehyde Removal through Activated Carbon Adsorption

Yu-Jin Kang [1], Hyung-Kun Jo [1], Min-Hyeok Jang [1], Xiaoliang Ma [2], Yukwon Jeon [3], Kyeongseok Oh [4,*] and Joo-Il Park [1,*]

1 Department of Chemical & Biological Engineering, Hanbat National University, Daejeon 34158, Korea; yujin0581@naver.com (Y.-J.K.); 30211222@edu.hanbat.ac.kr (H.-K.J.); 30211219@edu.hanbat.ac.kr (M.-H.J.)
2 Petroleum Research Center, Kuwait Institute for Scientific Research, P.O. Box 24885, Safat 13109, Kuwait; maxiaoliang@kisr.edu.kw
3 Department of Environmental and Energy Engineering, Yonsei University, Wonju 26493, Korea; ykjeon@yonsei.ac.kr
4 Department of Chemical & Biological Engineering, Inha Technical College, Incheon 22212, Korea
* Correspondence: kyeongseok.oh@inhatc.ac.kr (K.O.); jipark94@hanbat.ac.kr (J.-I.P.); Tel.: +82-32-8702271 (K.O.); +82-42-821153 (J.-I.P.)

**Abstract:** Formaldehyde is a highly toxic indoor pollutant that can adversely impact human health. Various technologies have been intensively evaluated to remove formaldehyde from an indoor atmospheres. Activated carbon (AC) has been used to adsorb formaldehyde from the indoor atmosphere, which has been commercially viable owing to its low operational costs. AC has a high adsorption affinity due to its high surface area. In addition, applications of AC may be diversified by the surface modification. Among the different surface modifications for AC, amination treatments of AC have been reported and evaluated. Specifically, the amine functional groups of the amine-treated AC have been found to play an important role in the adsorption of formaldehyde. Surface modifications of AC by impregnating and/or grafting the amine functional groups onto the AC surface have been reported in the literature. The impregnation of the amine-containing species on AC is mainly achieved by physical interaction or H-bond of the amines to the AC surface. Meanwhile, the grafting of the amine functional groups is mainly conducted through chemical reactions occurring between the amines and the AC surface. Herein, the carboxyl group, as a representative functional group for grafting on the surface of AC, plays a key role in the amination reactions. A qualitative comparison of amination chemicals for the surface modification of AC has also been discussed. Thermodynamics and kinetics for adsorption of formaldehyde on AC are firstly reviewed in this paper, and then the major factors affecting the adsorptive removal of formaldehyde over AC are highlighted and discussed in terms of humidity and temperature. In addition, new strategies for amination, as well as the physical modification option for AC application, are proposed and discussed in terms of safety and processability.

**Keywords:** activated carbon; formaldehyde removal; amination; surface modification; functional groups

## 1. Introduction

Exposure to high levels of pollutants and hazardous gases adversely affects human health, and a significant challenge is to mitigate the adverse effects by reducing the levels of the pollutants and hazardous gases in areas of human habitation. In particular, toxic volatile organic compounds (VOCs) pose a grave threat to human health, and prolonged exposure to VOCs can result in a number of adverse effects on human health [1–3]. Formaldehyde is a major indoor VOC and is classified as a group 1 carcinogen according to the WHO International Agency for Research on Cancer (IARC) [4]. Furthermore, it has been reported that formaldehyde may cause irritation of the eye, nose, skin, throat, or combinations thereof. Moreover, long-term exposure to formaldehyde can be linked to asthma, sinonasal

disorders, and even some cancers [5]. Formaldehyde is present in many objects, such as building materials, home interior decorative materials, and burning instruments [1,4], and may be emitted from the objects into their surroundings. Due to increased construction of structures such as bridges, roadways, buildings, and other infrastructure, the use of building materials has correspondingly increased, leading to an overall increase in levels of formaldehyde present in the environment.

Many countries have regulated lower levels of formaldehyde concentrations in indoor air, including as low as two-digit ppb(ug/m$^3$) levels [6]. To conform to such regulations, numerous approaches have been used to improve the effectiveness of the removal of formaldehyde from the air, including its recovery and destruction. Methods for the recovery of formaldehyde include physisorption and chemisorption, and methods for the destruction of formaldehyde include thermal and photocatalytic oxidation. The merits/demerits of each of the methods for formaldehyde removal are presented in Table 1. Among the methods, thermal oxidation requires high energy to substantially oxidize formaldehyde (HCHO) to carbon dioxide ($CO_2$) and water ($H_2O$). Photocatalytic oxidation of formaldehyde may require noble metals as catalysts, thereby making the process expensive. Further, photocatalysis of formaldehyde may generate undesirable byproducts. As alternates, non-thermal plasma and biomaterial-filtration methods have been proposed to reduce the concentration of formaldehyde in the air [7,8]. Non-thermal plasma can efficiently oxidize low concentrations of formaldehyde. However, the process may result in the generation of undesirable byproducts, such as ozone and NOx. Furthermore, while biomaterial filtration has been proposed as an eco-friendly method to reduce the concentration of formaldehyde, the methodology of biomaterial filtration is still required development of a design for the effective microbial consumption of formaldehyde [9].

**Table 1.** Comparison of formaldehyde removal options.

| Methods | Technology | Concentration | Operating Temp. | Operating Cost |
|---|---|---|---|---|
| Recovery | Adsorption | Low, High | Ambient | Acceptable |
| | Membrane | High | Ambient | High cost of material |
| | Condensation | High | Ambient and cryogenic | High cost of energy |
| Destruction | Thermal Oxidation | High | ~815 °C | High cost of Energy |
| | Catalysis | Low, High | Ambient, 200~500 °C | Acceptable |
| | Photo catalysis | Low, High | Ambient | Costly dopants required |
| | Non-thermal plasma w/wo catalyst | Low | Ambient | High cost of system assembly |
| | Biological/Botanical filtration | Low | - | - |

Some unconventional methods, such as membrane and condensation methods, have been explored and have been demonstrated to remove formaldehyde when it occurs at low concentrations. However, such methods are very expensive. In contrast, the adsorption methods offer attractive alternatives for reducing formaldehyde owing to its simplicity, ease of operation, and low operational cost [3,5,10–19]. Various adsorbents have been studied to regulate the indoor VOC content. Recently, some adsorbents with homogeneous nano-characteristics have been presented that have shown superior adsorption performance compared to that of conventional adsorbents [20]. However, this special nano-adsorbent requires expensive handling. Therefore, the modification of conventional adsorbents, especially activated carbon (AC), is considered to achieve efficient formaldehyde removal in practical applications. In general, AC shows effective adsorption of VOCs with boiling points above 0 °C but is inefficient when the boiling points of VOCs is lower than 0 °C; in these cases, the physical adsorption is dominant. The boiling point of formaldehyde is −19.5 °C. To improve the adsorption efficiency, AC surface modification is often employed. Surface modification allows the physical and chemical characteristics of the AC surface to enable effective adsorption of formaldehyde. The adsorption capabilities of formaldehyde appear complicated in the presence of moisture. The adsorption competition between water and formaldehyde on the surface of activated carbons seems indispensable. The chemical

modification of the AC surface, especially through introducing amine functional groups onto the surface, is reviewed and highlighted. In addition, the physical fabrication option followed by the safe handling of amination options for AC applications are proposed and discussed in Section 5. We hypothesize that a better performance can be achieved through well-designed activated carbons, such as high surface area, well-distributed micropores, and mechanical hardness. In the near future, the authors are expecting to evaluate our customized polymer-based activated carbons with previously fabricated morphologies combined with surface modification options. In brief, the surface-modified bead-type AC by amination is highly promising for practical application in removing formaldehyde from indoor air. The suggested activated carbon is not a granular activated carbon but a bead-type activated carbon, which would be applicable to a convenient recycle process operated in a flow-loop system. However, the viability of commercialization should be verified.

This paper briefly reviews thermodynamics and kinetics for adsorption of formaldehyde on AC and the current progress in surface modification of AC by amination for adsorptive removal of formaldehyde from an indoor atmosphere. Various physical (operational) and chemical factors that may affect the effective adsorption of formaldehyde are presented and discussed. In Sections 3 and 4, the effect of amination of the AC surface is highlighted, and the adsorption mechanism of formaldehyde on AC and modified AC is discussed. In addition, strategies for the practical application of AC are proposed by sharing our recent work. In the Section 5, as mentioned above, two aspects of AC modification are suggested in terms of the usage of less toxic amines as well as physical modification options for favorable fluidization in the regenerability of spent adsorbent.

## 2. Thermodynamics and Kinetics for Adsorption of Formaldehyde on AC

The thermodynamics and kinetics for adsorption of formaldehyde on AC are the basis for discussing the physical and chemical factors that affect the adsorptive removal of formaldehyde, which will be discussed in Sections 3 and 4. Consequently, these aspects are briefly reviewed below.

### 2.1. Thermodynamics and Adsorption Isotherms

Thermodynamic analysis of formaldehyde adsorption on AC is important for determining whether adsorption or sorption is spontaneous or not. The value changes in standard enthalpy ($\Delta H^\circ$), standard entropy ($\Delta S^\circ$), and standard free energy ($\Delta G^\circ$) can provide evidence that the adsorption reaction is viable. The representative equations are shown below:

$$\ln K_d = \frac{\Delta S^\circ}{R} - \frac{\Delta H^\circ}{RT} \tag{1}$$

$$K_d = \frac{C_{AB}}{C_B} \tag{2}$$

$$\Delta G^\circ = -RT \ln K_d \tag{3}$$

where $T$ is temperature; $K_d$ is adsorption adsorption/sorption equilibrium constant. It has been reported that the adsorption of formaldehyde on Ag-AC, CaO-AC, and $Fe_3O_4$-AC was exothermic ($\Delta H^\circ$ values was negative), and the negative values of $\Delta G^\circ$ indicate the feasible and spontaneous nature of their adsorption process [21,22]. The equations above also indicate that the adsorption capacity decreases with increasing adsorption temperature.

Meanwhile, adsorption isotherms provide important information about the adsorption capacity of the adsorbent as a function of adsorbate partial pressure in the gas phase. According to the literature [21–30], there are two major adsorption isotherm models describing formaldehyde behaviors on ACs (ACF or surface-modified ACs): the Langmuir model and the Freundlich model. These models depend on the surface properties, the pore structure of adsorbents, and adsorption conditions. In general, the Langmuir isotherm model is based on the ideally single-layer adsorption on the surface of the adsorbent. On the other hand, the Freundlich isotherm model is an empirical equation applicable to a

heterogeneous surface and multi-layer adsorptions. Both cases also depend on the concentrations of adsorbate. For nonlinear isotherms, the Qi-LeVan model has been proposed [23]. Representative isotherm equations are shown below, in the order of Langmuir, Freundlich, and Qi-LeVan models.

$$q_e = \frac{q_m \, K_c \, C_e}{1 + K_L \, C_e} \tag{4}$$

$$q_e = k_f C_e^{1/n} \tag{5}$$

$$p = \frac{n}{\varepsilon_0 + \varepsilon_1 n + \varepsilon_2 n^2 + \varepsilon_3 n^3} \tag{6}$$

where $C_e$ is the formaldehyde concentration at the equilibrium, $q_e$ is the formaldehyde amount adsorbed on adsorbents, $q_m$ is the maximum adsorption amount of formaldehyde on adsorbents, which is recognized as the Langmuir's parameter, and $K_L$ is the Langmuir's adsorption equilibrium constant. In Equation (6), $p$ is the partial pressure, n is the surface loading, and $\varepsilon_i$ is defined as the adsorbent–adsorbate pair parameter [23].

In Table 2, the adsorption isotherm models for formaldehyde on surface-modified AC are listed. It has been reported that the Langmuir model is more suitable than the Freundlich model when formaldehyde adsorbs on the surface of modified ACs ($MnO_2$ [24], potassium [25], EDA [26], and Ag [27]) and AC adsorptive filter media [28]. In these cases, a linear regression coefficient ($R^2$) is often used to determine the best fit for isotherm models [24–28]. In addition, Chang et al. [29] and Rengga et al. [30] also found that the Langmuir model is suitable for describing the adsorption isotherm of formaldehyde on the Ag-modified ACs. The reason why the Langmuir model showed better fit can be attributed to the hypothesis that the surface-modified ACs interact with formaldehyde by chemisorption through their functional groups on the surface and result in single-layer molecular adsorption [21,24–29].

**Table 2.** Adsorption isotherms for formaldehyde on surface-modified AC as reported in the literature.

| Modification AC | Operating Conditions: T (°C)/Humidity (%)/pH | Ce, Feed Concentration (ppm) of FA Up to | Qe (mg/g) | Ce vs. Qe, Linear/Nonlinear (Isotherm Model) | Ref. |
|---|---|---|---|---|---|
| Ag-AC | 25, 35, 45, 60/-/- | 1000 | <120 | Nonlinear (Langmuir) | [21] |
| CaO-AC Fe₃O₄-AC | 25/-/2–10 | 10 | <20 | Nonlinear (Freundlich) | [22] |
| Granular AC and ACF | 26/-/- | 35 | <450 | Nonlinear (Qi-LeVan) | [23] |
| MnO₂-AC | 25/-/- | 0.2 | <0.1 | Nonlinear (Langmuir) | [24] |
| Potassium-AC | 28 ± 2/40 ± 2/- | 0.9 | <0.4 | Nonlinear (Freundlich and Langmuir) | [25] |
| * EDA-AC | 25/-/- | 2.45 | <2.5 | Nonlinear (Langmuir) | [26] |
| Ag-AC | 25/-/- | 14 | <7 | Linear (Langmuir and Freundlich) | [27] |
| AC adsorptive filter media | 28 ± 2/40 ± 2/- | 0.8 | <0.75 | Nonlinear (Langmuir) | [28] |
| Ag-AC | Room/-/- | 0.9 | <35 | Linear (Langmuir) | [29] |
| Ag-AC, Cu-AC | 25/-/- | 20 | <0.6 | Linear (Langmuir and Freundlich) | [30] |

* EDA: Ethylenediamine.

Meanwhile, isotherm models have been evaluated in the literature. Khaleghi et al. [22] investigated the adsorption of formaldehyde over AC-decorated CaO and $Fe_3O_4$ nanoparticles, and adsorption behaviors were evaluated using the three isotherms of the Langmuir model, Freundlich model, and Dubinin-Radushkevich model. They found that the Freundlich model was the best fit after comparing the $R^2$ values of all three models. They concluded that their adsorption process was a physical and multilayer adsorption. In other respects, Carter et al. [23] examined the adsorption behaviors using commercialized activated carbon fiber cloth and granular ACs, and they reported that the adsorptions follow the Type V isotherm, which is different from the normal Type I behavior [23]. Thus, they proposed the Qi-LeVan model as the best fit. They also suggested the following three factors to explain Type V behavior: (1) reduction to a finite, nonzero limit at low partial pressures, (2) description of the adsorption isotherm over the entire range of adsorption considered, and (3) successful fit of both HCHO and water vapor adsorption isotherms.

### 2.2. Kinetic Models

Many researchers have reported diverse adsorption kinetic models for formaldehyde, such as the pseudo-first-order model (Lagergren equation), the pseudo-second-order model (HO and McKay equation) and the Bangham model (Bangham equation) [22,25–28,31].

Kinetic models for the adsorption of formaldehyde on surface-modified AC are presented in Table 3. Representative equations are shown below in the order of pseudo-first-order, pseudo-second-order, and Bangham model.

$$\frac{dq_t}{dt} = k_2 (q_e - q_t)^2 \tag{7}$$

$$\log(q_e - q_t) = \log q_e - \frac{k_1}{2.303} t \tag{8}$$

$$q_t = q_e - \frac{q_e}{e^{k_B f^z}} \tag{9}$$

where $q_e$ ad $q_t$ are the amount of adsorbate uptake per mass unit of adsorbent at equilibrium and at a certain time $t$, respectively; $k_i$ is the rate constant; $k_B$ and $z$ are Bangham parameters. According to kinetic equations (Equations (7)–(9)), sorption capacity ($q_e$ and $q_t$), adsorption rate constant ($k_1$ and $k_2$), and correlation coefficient (often referred to as $R^2$ value) can be obtained by fitting experimental data. In specific, when the surfaces of ACs were modified by PEI [31], potassium [25], Ag [27], CaO, and $Fe_3O_4$ [30], pseudo-second-order kinetic models showed the best fit. On the other hand, the Bangham model was the best fit for ethylenediamine-modified ACs [26].

**Table 3.** Kinetic models for adsorption of formaldehyde on surface-modified AC reported in the literature.

| Modification AC | Adapted Model | Concentration (ppm) | Ref. |
|---|---|---|---|
| CaO-AC Fe₃O₄-AC | Pseudo-second-order | 5–50 | [22] |
| Potassium-AC | Pseudo-second-order | 0.25, 0.5, 0.75, 0.9 | [25] |
| EDA-AC | Pseudo-first-order Bangham | 2.45, 8.15 | [26] |
| Ag-AC | Pseudo-second-order | 0~14 | [27] |
| AC adsorptive filter media | Pseudo-second-order | 0.25, 0.56, 0.79 | [28] |
| * PEI-AC | Pseudo-second-order | 50 | [31] |

* PEI: Polyethyleneimine.

Therefore, various factors should be evaluated in kinetic models of formaldehyde adsorption. Various factors include operational temperature, partial pressure, the amount of adsorbents, and so on. In many cases, chemisorption and physisorption occur at the same time. The surface functional group will contribute more chemisorption, but physisorption

is still possible as long as adsorptive surfaces of activated carbon participate. Adsorption is affected by pore size and pore volume. As the adsorption proceeds, the inlet of pores may be blocked and may inhibit the adsorbate to diffuse within the inner pores. In this case, physisorption kinetics may be unfavorably affected, and the delay or the prevention of adsorption will also negatively affect overall adsorption kinetics. Meanwhile, pseudo-first-order adsorption may occur when very low concentration of adsorbate are applied. It can be expected that the increase of adsorbate will result in the linear increase of overall adsorption in low concentration ranges.

## 3. Physical (Operational) Influence Factors

The adsorption behavior of AC is classified according to chemisorption and physisorption. Chemisorption occurs through chemical bonding between the chemical and adsorbent, while physisorption occurs through bonding of the adsorbate on the surface of the adsorbent due to weak van der Waals forces and capillary interactions. Typically, AC is a porous material with abundant micropores, and thus AC has a high surface area. AC may be produced from myriad sources, such as fossil fuel residues and various types of biomass. Furthermore, coconut shells and charcoal are often considered as common examples of activated carbon materials [32,33].

In studies on the adsorption of formaldehyde (FA) on AC under humid conditions, three different pathways of adsorption of FA on AC have been suggested in the presence of water vapor [34]. First, the competitive adsorption of FA with $H_2O$ due to their similar polarities, which takes place on active sites located on the pore surface of the AC, can have a negative effect on FA removal. Recently, the FA removal efficiency of nitrogen- and sulfur-modified AC was evaluated under humid conditions [35]. Even though the modified AC showed an overall improvement in efficiency of FA removal, the presence of water vapor resulted in a decrease in adsorption capacity over all adsorbents (ca. 3–40% decrease, based on FA (mg)/g adsorbent). Interestingly, it was found that the introduction of organosilane onto the AC surface improved the adsorption performance for the adsorption of FA over a wide range of humidities [36,37]. In fact, the FA removal efficiency of the organosilane-modified AC increased to 48% at a relative humidity (RH) of 80%, while the FA removal efficiency of pristine AC decreased by 86% at a relative humidity of 80% when compared to that at RH 30%. Other studies have also presented good examples of competitive adsorption of FA with $H_2O$ [38–43]. Lee et al. [12] reported that an increase in the water affinity of adsorbents could be obtained by introducing polar functional groups. However, this may lead to a considerable decrease in the adsorption capacity of the AC for FA due to the adsorbents with preferential adsorption of water. Therefore, an optimized amount of polar functional groups should be considered for FA adsorption by reducing water affinity.

A second pathway for adsorption of FA on AC suggests that the exposed surface area may be reduced by the capillary condensation of water molecules within the micropores of the AC, which may further reduce the adsorption capacity of AC. Pei et al. reported a higher capillary condensation of water vapor within the micropores of AC under higher RH conditions, resulting in a decrease in the FA adsorption capacity [11]. It was indicated that nano-sized pores with less than 2.4 nm diameter were unfavorably blocked due to capillary condensation at RH 80%. Similar results also demonstrated that increase in relative humidity decreased the FA breakthrough capacity in the case of nitrogen-modified AC [44].

A third pathway for the adsorption of FA on AC suggests that the FA molecules can be co-absorbed by the condensed and/or adsorbed water on the surface. If this is true, the FA removal efficiency may increase because the boiling point of FA is much lower than that of water vapor, which is more difficult to adsorb. The co-adsorption of FA and water has a positive effect on FA removal, whereas the opposite is true for the first and second pathways mentioned above. Therefore, the phenomenon of FA adsorption remains contentious.

The authors of the present manuscript have attempted to improve the adsorption performance of bead-type AC (BAC) and two acid-modified BACs (sulfuric acid-modified BAC (SA-BAC) and hydrochloric acid-modified BAC (HA-BAC)) at different RH values. The results are shown in Figure 1. The adsorption data (mg-FA/g-AC) showed a similar trend in all three cases: a higher capacity at a higher RH (70%) and a lower capacity at a lower RH (30%). FA adsorption clearly improved at RH 70% in all three cases. Notably, both SA-BAC and HA-BAC showed a greater improvement in FA adsorption capacities than BAC at RH 70% (Figure 1). These results support the claim that the absorption of FA into the adsorbed water on the pore surface may be enhanced when surface modifications are applied.

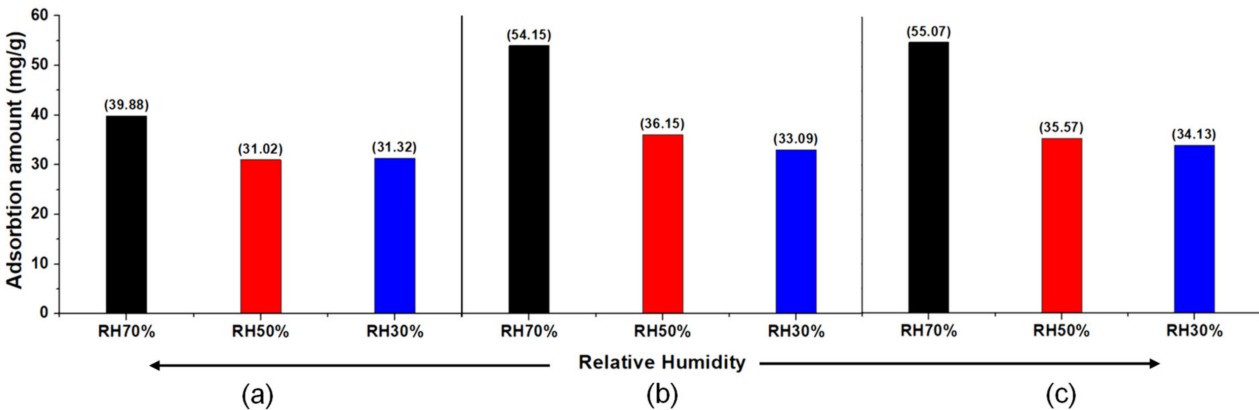

**Figure 1.** FA removal efficiency on the (modified) bead-type activated carbon at the various RH contents: (**a**) non-modified bead-type activated carbon, (**b**) halogen modified bead-type activated carbon, and (**c**) sulfur modified bead-type activated carbon. Operation conditions: 100 ppmv; space velocity: 40,000/h; temperature: ambient.

Temperature should also be considered a significant factor for FA adsorption on AC, as indicated in Section 2.1. Generally, the adsorption process can occur at room temperature, unlike the thermal or catalytic oxidation process, which requires elevated temperatures. Many studies have suggested that a strong interaction between the AC surface and FA molecules promotes removal capacity [21,45–49]. In addition, as an increase in temperature can increase the entropy on the AC surface, FA molecules are easily desorbed, because adsorption is an exothermic process (see Section 2.1). This explains the decrease in the adsorption capacity with increasing temperature.

## 4. Approaches to Chemical Modifications of the AC Surface

Generally, carbonaceous materials exhibit a good adsorptive performance and mechanical stability. Apart from physical properties such as surface area and pore characteristics, surface functional groups that are used for surface modification of AC play an important role in the adsorption capacity of AC [50,51]. When we consider the direct FA adsorption on the surface of AC, the hydrophobic surface of AC is not favorable. However, surface modification of AC can enable the AC to efficiently adsorb hydrophilic FA. Therefore, the adsorptive performance of AC may be enhanced by surface modification, which may be achieved through three means: acid or base treatment, impregnation with hydrophilic chemicals, and grafting of hydrophilic functional groups connected to the AC surface [52,53].

Considering the first option, modification to brace acidic or basic functional groups on the AC surface has been extensively evaluated [16,31,54–58]. Generally, the surface chemistry of AC depends on the presence of O, N, S, and H within the functional groups on its surface. Returning to Figure 1, the results support the evidence that sulfur- or halogen-modified BAC (bead-type AC) has better adsorption capacity than pristine BAC. When

we consider effective FA removal, the oxygen functional groups of the Brønsted acidic or basic sites, as well as nitrogen-containing functional groups, can be the most important sites for FA adsorption. This point is also supported by the adsorption isotherms of the Langmuir model, which was found suitable for the adsorption of FA on surface-modified ACs. Hence, modifying the surface functional groups of AC is a key strategy for designing the surface chemistry.

Considering second option, nitrogen-containing functional groups can be introduced as dopants in the AC framework with heat treatment in the presence of nitrogen sources (such as $NH_3$, $HNO_3$, etc.). The term dopant used here refers to the active chemical that is impregnated into the AC surface structure and forms physical bonds: Van der Waals and hydrogen bonds. Figure 2 shows the conceptual impregnation of aniline into the AC surface structure. A similar conceptual drawing can be seen elsewhere [59]. These physical bonds between aniline and the AC surface may be much weaker than the chemical bonds, but these bonds are still sufficient to hold dopants within the AC surface structure without detaching during the adsorption process [60]. The dopant was safely located within the AC surface structure with basic characteristics and surface polarity ($\pi$-electrons). Other nitrogen-containing functional groups include -NH$_2$, -NH, -C=N, and -C-N. Notably, -NH$_2$ (amine functional group) is attributed to the amine-aldehyde conjugation through covalent bonding and shows an enhanced adsorption capacity for FA [60]. Such nitrogen-containing functional groups have a strong affinity for water molecules on the adsorbent [20]. In addition, oxidation and sulfurization can successfully modify the AC surface by increasing the functional groups, such as -OH, -COOH, -C=O, -C-O, -C-S, -C=S, and -S=O, on the AC surface.

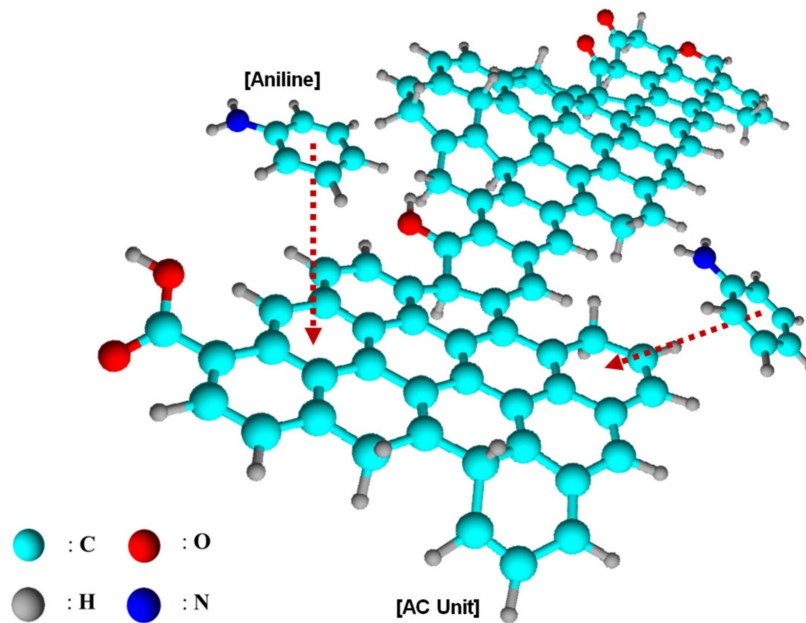

**Figure 2.** Conceptual diagram of impregnated aniline on the surface of activated carbon.

Considering the third option, the grafting of hydrophilic functional groups can be achieved by a direct reaction between a hydrophilic reactant and intrinsic hydrophilic functional groups on the AC surface. Two-stage reactions can also be applied to build grafting structures. For example, carboxylic functional groups can be used to explain the generation of a grafting structure. If intrinsic carboxylic groups are present on the AC surface, a condensation reaction occurs between the carboxylic group and the supply of hydrophilic chemicals such as amines. In this study, amination was used to describe the grafting reaction. Figure 3 presents two examples of reactions that form grafting structures [61,62]. Figure 3a shows the condensation reaction between melamine and

carboxylic functional groups on the AC surface. Figure 3b shows the reaction between diethyl triamine and a carboxylic group. Both grafting reactions proceed via halogenated intermediates. If there are fewer carboxylic groups on the AC surface, acid treatment using $HNO_3$ can generate carboxylic groups, followed by amination for grafting [50]. Table 4 summarizes the representative amination chemicals and amination reaction procedures. These aminated ACs showed a greater improvement in the adsorption performance in various adsorption processes.

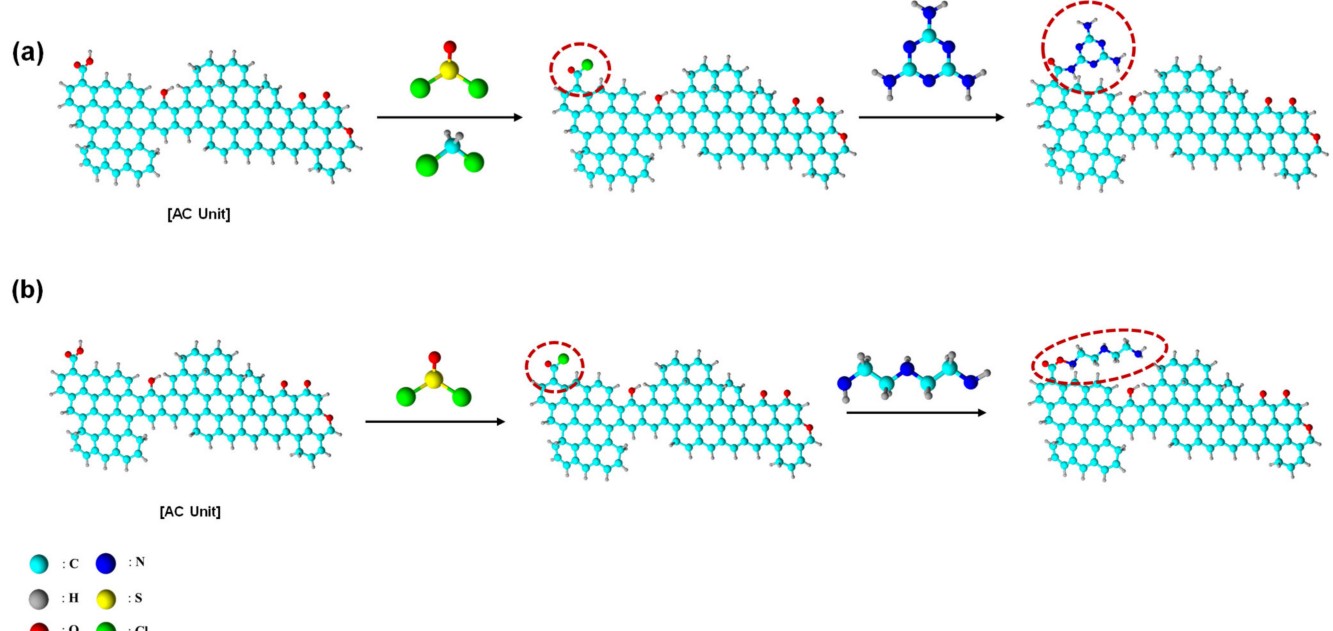

**Figure 3.** Grafting reactions (**a**) between melamine and activated carbon and (**b**) between diethylene triamine and activated carbon via halogenated intermediates for both cases.

The mechanism of FA is described in our previous report [63]. Considering urea as an amine source, the carbonyl group of FA adsorbs onto the active sites (micro-walls) of AC ($HCHO_{(g)} \rightarrow HCHO_{(ad.)}$), in which the adsorbed FA, presenting $\delta^+$, nucleophilically reacts with the amine group to form an imine ($HCHO_{(ad.)} + R\text{-}NH_2 \rightarrow R\text{-}N=CH_2 + H_2O$). In parallel, adsorbed FA can be oxidized to produce formic acid ($HCHO_{(ad.)} + 1/2O_2 \rightarrow HCOOH_{(ad.)}$). Formic acid reacts with imine to produce a secondary amine ($HCOOH_{(ad.)} + R\text{-}N=CH_2 \rightarrow R\text{-}NHCH_3 + CO_2$). Meanwhile, the absorbed formic acid can be dissociated to $H^+$ and $HCOO^-$, where $H^+$ may react with the secondary amine and the adsorbed FA to produce iminium species ($H^+ + HCHO_{(ad.)} + R\text{-}NHCH_3 \rightarrow R\text{-}N+CH_3CH_2 + H_2O$). Subsequently, the reaction of the iminium species and $HCOO^-$ leads to the formation of 1,1-dimethylurea ($R\text{-}N+CH_3CH_2 + HCOO^- \rightarrow R\text{-}N(CH_3)_2 + CO_2$). The concept of the mechanism of FA removal through amination of AC is illustrated in Figure 4.

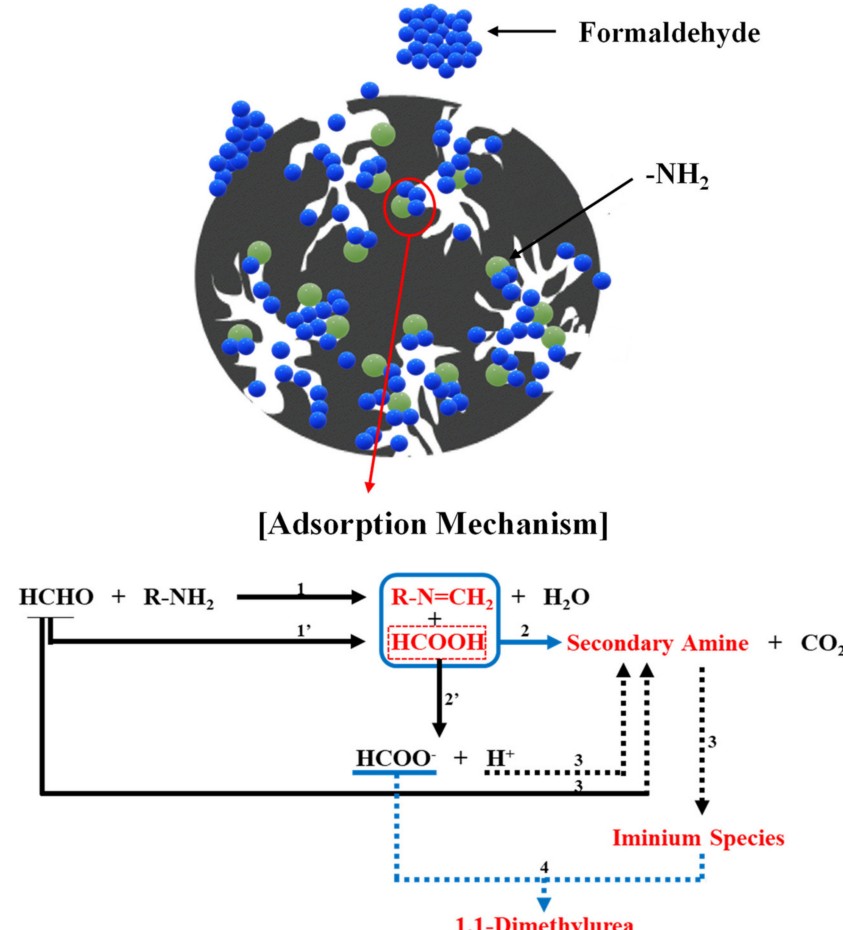

Figure 4. The mechanism of FA over amine species (urea) onto the AC (modification from our previous work [63]).

**Table 4.** Interaction states of amine chemicals after amination treatment of activated carbon and their applications.

| Chemical Structure | Interaction with AC | Application | Ref. |
|---|---|---|---|
| Polyethyleneimine (PEI) | Physical bonding | Formaldehyde removal | [31] |
| Hexamethylene diamine (HMDA) | Physical bonding and grafting; it was reported that the chemical reaction between HMDA and AC may be dominant | Formaldehyde removal | [57] |
| Urea | Physical bonding; formaldehyde and nitric acid were co-impregnated | Formaldehyde removal | [60] |

**Table 4.** *Cont.*

| Chemical Structure | Interaction with AC | Application | Ref. |
|---|---|---|---|
| Melamine | Grafting; melamine was used as an intermediate to grow structure with covalent organic polymer | Organic dyes and metals adsorption | [62] |
| Diethylene Triamine (DETA) | Grafting | Phenol removal in aqueous phase | [62] |
| | Physical bonding | Formaldehyde removal | [64] |
| P-aminobenzonic acid (PABA) | Physical bonding (deposition) | Formaldehyde removal | [65] |
| Etilenodiamina (ethylenediamine) | Grafting after nitric acid treatment | Formaldehyde removal | [66] |

## 5. Challenge to Industrial Application

Formaldehyde is a highly toxic indoor pollutant that poses a grave threat to human health. Various technologies have been intensively evaluated for the effective removal of formaldehyde from indoor air. AC has a relatively high adsorption affinity, owing to its high surface area. The diverse commercialization of activated carbon (AC) has motivated a review of current technologies for formaldehyde removal in terms of AC surface modification. Among these modifications, amination is widely used to remove low concentrations of formaldehyde. In the literature, aminations are often achieved by impregnation and/or grafting of amine functional groups onto the AC surface. The amine functional groups participated in the adsorption of formaldehyde. In the case of grafting, the amine functional group reacts to form chemical bonds with intrinsic functional groups on the AC surface. Herein, the carboxyl group, as a representative functional group on the surface of AC, was used to demonstrate amination reactions with various amine chemicals. When considering the commercial application of AC, the outcome of toxic chemicals must be considered. In particular, amination reactions are induced by toxic amine compounds. Therefore, less toxic amine sources are necessary and are recommended. We classified amine sources based on toxicity/nontoxicity (Table 5). Moreover, the application of AC in industry has always been limited by safe working environments. Representative issues include dust (explosion) and handling feasibility. To overcome these problems, new types of AC are required. For example, AC with higher compressive strength and fluidity is recommended. In this regard, the present authors developed a new bead-type (spherical) AC with a high surface area (exceeding 1500 m$^2$/g), high compressive strength (9 kg$_f$/unit), and fluidity with a mean particle size of less than 1 mm. AC with these physical properties is expected to be a safer alternative to conventional AC. In particular, the application of this bead-type AC will benefit industries with high FA emissions, such as the wood (furniture) processing industry. In addition, it can be effectively used as the purifier of an indoor environment, with processability owing to the fluidity of the bead-type AC. This type of AC can be easily operated for adsorption and desorption, thereby making it feasible for recycling (Figure 5). More details regarding customized BAC are still in the evaluation stage and will be provided in future studies.

**Table 5.** Amine sources based on toxic/non-toxic chemicals.

| Chemicals | Toxic/Non-Toxic | Ref. |
|---|---|---|
| Polyethyleneimine | Toxic | [31] |
| Dicyandiamide | Toxic | [42] |
| Thiourea | Toxic | [42] |
| Penicillin G | Toxic | [42] |
| Nitric Acid/Sulfuric Acid | Toxic | [50] |
| Hexamethylene Diamine | Toxic | [57] |
| Urea | Non-toxic | [60] |
| Melamine | Non-toxic | [62] |
| Diethylene triamine | Toxic | [64] |
| P-aminobenzoic acid | Toxic | [65] |
| Etilenodiamina | Toxic | [66] |
| Hexamethylenetetramine | Toxic | [67] |
| 3-aminopropyltriethoxysilane | Toxic | [68] |
| Diallylamine | Toxic | [69] |
| Piperazine | Non-toxic | [70] |

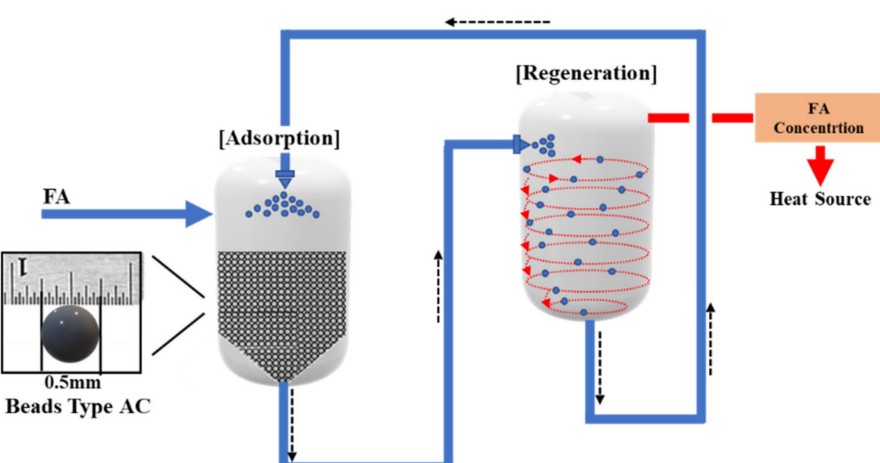

**Figure 5.** Conceptual diagram of recycling process for customized bead-type AC with 0.5 mm diameter, on average.

## 6. Summary

Formaldehyde removal is important to maintain clean air in indoor environments. Among the formaldehyde removal options, activated carbon adsorption is attractive because of its economic application. Lately, diverse origins of activated carbon materials have been explored to evaluate whether a customized activated carbon performs efficiently in the removal of emerging contaminants. Activated carbon has been examined in various applications, such as wastewater treatment and gas-phase adsorption. The present paper briefly reviewed the thermodynamics and kinetics for the adsorption of formaldehyde on AC, as well as the current progress in surface modification of AC by amination. The major factors affecting the adsorptive removal of formaldehyde over AC are presented and discussed in terms of humidity and temperature. The adsorption capabilities of formaldehyde seem complicated in the presence of moisture. The competition between water adsorption and formaldehyde adsorption on the surface of activated carbons seems to be a complicating factor. The chemical modification of the AC surface, especially through introducing amine functional groups onto the surface, to improve the adsorption capacity is reviewed and highlighted. The surface modification of AC by amination is an important strategy to increase the adsorption capacity for removing formaldehyde. In addition, the physical fabrication and safe handling options for amination in AC applications were proposed and discussed. Better performance can be achieved from well-designed activated carbons that have high surface area, well-distributed micropores, and mechanical hardness. The

authors are preparing and evaluating customized polymer-based activated carbons with previously fabricated morphologies combined with surface modification options. The performance results will be presented in future studies. The surface-modified bead-type AC by amination is highly promising for practical application in removing formaldehyde from indoor air. A suggested activated carbon called the bead-type activated carbon, which is not a granular activated carbon, could be part of a convenient recycling process, because it conveys used and recycled activated carbons in a flow-loop system. However, it should be verified whether commercialization is viable.

**Author Contributions:** Conceptualization, methodology and writing (original draft, review and editing): K.O. and J.-I.P.; supervision and funding acquisition: J.-I.P.; experiment performance, data processing, software and formal analysis: Y.-J.K., H.-K.J., M.-H.J., X.M. and Y.J. All authors have read and agreed to the published version of the manuscript.

**Funding:** This work was supported by Korea Environment Industry & Technology Institute (KEITI) through the Prospective Green Technology Innovation Project, funded by Korea Ministry of Environment (MOE) (2020003160004).

**Conflicts of Interest:** The authors declare no conflict of interest.

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
