# Peer review of "A Brief Review of Formaldehyde Removal through Activated Carbon Adsorption"

_applsci, doi:10.3390/app12105025_

Round 1

Reviewer 1 Report

I regret to state that the manuscript presented for peer review does not meet the standards of the review article. The title promises an overview in the subject “A Brief Review of Formaldehyde Removal through Activated Carbon Adsorption”. Unfortunately, the content of the article is significantly limited and does not cover all topics related to the methods of removing formaldehyde with the use of activated carbon. In the approach “Removal of formaldehyde by adsorption of activated carbon”, it is possible to present your own solutions, provided that they are discussed in a broad context and critically discussed. This article lacks such an approach.

Other disadvantages of the article:

  • The literature review is very limited.
  • The purpose for which the review was carried out was not specified
  • 56-58: “A report of the United States Environmental Protection Agency (US-EPA) has mandated that concentration of formaldehyde in the air around newly constructed buildings be regulated to no more than 0.016 ppm [6]” - This source does not give such information. Formaldehyde concentrations in outdoor air are generally much lower.
  • What is revealing in Figure 2?
  • It is difficult to understand what is shown in Figure 3.
  • The mechanism presented in Figure 5 is incomprehensible.
  • It is not known what is discussed in the Discussion section
  • Figure 6 shows only the formaldehyde adsorption and desorption scheme and nothing else.
  • There are no final conclusions.

Author Response

Reviewer 1.

Comment:

I regret to state that the manuscript presented for peer review does not meet the standards of the review article. The title promises an overview in the subject “A Brief Review of Formaldehyde Removal through Activated Carbon Adsorption”. Unfortunately, the content of the article is significantly limited and does not cover all topics related to the methods of removing formaldehyde with the use of activated carbon. In the approach “Removal of formaldehyde by adsorption of activated carbon”, it is possible to present your own solutions, provided that they are discussed in a broad context and critically discussed. This article lacks such an approach.

Response: We revised the manuscript. First of all, adsorption isotherm studies have been added in the paragraph as well as new Table 2. The discussion regarding mechanisms of adsorption behaviors were also added. The figures were mainly revised for better resolution.

Comment:

Other disadvantages of the article:

  • The literature review is very limited.

Response: This article is the brief review of formaldehyde removal using surface modified activated carbon, mainly focused on the aminations. There are vast literature regarding activated carbons, for instance, in terms of preparations, adsorption performances, and isotherm modeling including adsorption kinetics. In fact, it is true that only limited applications are possible in commercial success in activated carbon adsorption. One of major objectives in this work includes the introduction of new application in commercial purpose. The commercial activated carbons are barely coming in air purifier area. The needs for better indoor qualities seems indispensable nowadays. We are trying to cover the options that may be future applicable to commercialization.

  • The purpose for which the review was carried out was not specified

Response: The purpose in our prior manuscript was to summarize the current options of surface modification of activated carbon. In general, there are two major precursors to prepare activated carbon. Biomass-based and synthetic material-based activated carbons were often referred. In our research lab, we are currently testing synthetic-based activated carbon that owns the high surface area. The characteristics of customized synthetic-based activated carbon will be prepared for a research article in near future. The purpose of this study were briefly written in the last paragraph in introduction part. We are more interested in the surface modification rather than material origins such as biomass or synthetic materials as mentioned earlier.

  • 56-58: “A report of the United States Environmental Protection Agency (US-EPA) has mandated that concentration of formaldehyde in the air around newly constructed buildings be regulated to no more than 0.016 ppm [6]” - This source does not give such information. Formaldehyde concentrations in outdoor air are generally much lower.

Response: We have changed the reference as the reviewer 1 pointed out above.

  • What is revealing in Figure 2?

Response: We intended to present the surface modification option through impregnating aniline inside the activated carbon structures without revealing detail chemical structure of activated carbon. It is definitely true that activated carbons have a lot of chemical functional groups depending upon their precursors’ origin. It is a schematic illustration to appeal that there is no chemical bonding between aniline and activated carbon. We added the explanation into Figure caption.

  • It is difficult to understand what is shown in Figure 3.

Response: The intention of Figure 3 is to show the physical bonding between amine chemical and activated carbon bearing surface functional group. The representative function group is carboxyl group. Even though there are many other functional groups on the surface of activated carbon as expected, the exemplary physical bonding such as Van der Waals was representatively presented. To avoid confusion, we decided to delete previous Figure 3.

  • The mechanism presented in Figure 5 is incomprehensible.

Response: Old Figure 5 has been replaced by new Figure 4 with the order of mechanism. It can be easily follow up the mechanism with ordering number of mechanism. It shows the better explanation regarding adsorption of formaldehyde and further decomposition to eco-friendly components. The serial reactions of adsorbed formaldehyde were previously presented in our previous article. The details have been already added in the manuscript.

  • It is not known what is discussed in the Discussion section

Response: We revised the discussion section. The structure of this article has been changed to firstly introducing thermodynamics and kinetics of formaldehyde adsorption. It should be noted that Isotherm kinetics were newly added in this revision. And physical influence and chemical modification were followed. In our discussion section, we suggest challenging approaches regarding the usage of less toxic amines and morphological challenges. The idea is to use polymer-based activated carbon that has much higher surface area. We would like to claim that the adsorption behavior will be more effective when surface modification was involved even no matter what material origins were selected for activated carbon. We focus on the importance of surface modification. In addition, we challenge to propose pre-shaped options (bead-type) using synthetic-based activated carbon. We expect the better operation with bead-type activated carbon because it also provides easier recycle treatment. The work relating to bead-type activated carbon is not the major scope of this review article but the idea suggesting in discussion.

  • Figure 6 shows only the formaldehyde adsorption and desorption scheme and nothing else.

Response: More detail drawing (Figure 5) replaced old Figure 6.

  • There are no final conclusions.

Response: As reviewer’s comment, we added the “Summary” section in the end of the manuscript.

Reviewer 2 Report

Reviewers Comments

Manuscript ID:  applsci-1650511

Title A Brief Review of Formaldehyde Removal through Activated Carbon Adsorption

Journal: Applied Sciences

General comments

The article reports review on AC capture of toxic indoor formaldehyde following a brief overview of various technologies that are used for its removal formaldehyde from an indoor atmosphere. The article discusses methods for AC surface modifications and how they influence the surface functional groups and also discusses some factors affecting the adsorptive removal of formaldehyde and proposed strategies for AC modification for application in formaldehyde indoor uptake. The manuscript is quite interesting, with low similarity index of 16%. Though the manuscript is well written with good command of English and presents significant data and analysis, yet it lacks significant data presentation, characterizations, influence of operational parameters and formaldehyde adsorption mechanisms presentation and interpretations. A major revision is needed prior to the acceptance of manuscript as per the following comments provided below 

Specific comments

  1. Comprehensive reviews on kinetics, equilibrium, and thermodynamics models’ fittings parameters for formaldehyde indoors sorption and elucidate the ACs formaldehyde uptake mechanisms interpretations
  2. Influence of the
  3. Provide and discuss comparative assessment of different AC types and precursor materials to further support understanding and interpretation of the formaldehyde adsorption mechanism
  4. A more comprehensive review discussion is needed on formaldehyde indoors adsorption with relevant tables format comparison of different types of AC in relation to influence of the physical and chemical operational parameters on efficiency and maximum adsorption capacity of indoors formaldehyde uptake

Author Response

Reviewer 2.

General comments

The article reports review on AC capture of toxic indoor formaldehyde following a brief overview of various technologies that are used for its removal formaldehyde from an indoor atmosphere. The article discusses methods for AC surface modifications and how they influence the surface functional groups and also discusses some factors affecting the adsorptive removal of formaldehyde and proposed strategies for AC modification for application in formaldehyde indoor uptake. The manuscript is quite interesting, with low similarity index of 16%.

Though the manuscript is well written with good command of English and presents significant data and analysis, yet it lacks significant data presentation, characterizations, influence of operational parameters and formaldehyde adsorption mechanisms presentation and interpretations. A major revision is needed prior to the acceptance of manuscript as per the following comments provided below 

Response: The purpose in our prior manuscript was to summarize the current options of surface modification of activated carbon. In general, there are two major precursors to prepare activated carbon. Biomass-based and synthetic material-based activated carbons were often referred. In our research lab, we are currently testing synthetic-based activated carbon that owns the high surface area. The characteristics of customized synthetic-based activated carbon will be prepared for a research article in near future. The purpose of this study were briefly written in the last paragraph in introduction part. We are more interested in the surface modification rather than material origins such as biomass or synthetic materials as mentioned earlier.

Specific comments

  1. Comprehensive reviews on kinetics, equilibrium, and thermodynamics models’ fittings parameters for formaldehyde indoors sorption and elucidate the ACs formaldehyde uptake mechanisms interpretations

Response: We summarized and added adsorption isotherm and kinetics in the section 2. Among the adsorption models, Langmuir and Freundlich isotherms were widely used in formaldehyde adsorption research. As we know, Langmuir isotherm is based on mono-layer adsorption and Freundlich is more empirical that is more applicable to heterogeneous surfaces and multi-layer adsorption. It is general that the indoor concentrations of formaldehyde may be low ppm level. If this is true, then Langmuir isotherm and first-order kinetics may have better chances to fit the data. Nevertheless, research works often explore wide ranges of formaldehyde concentrations. We favor to claim Langmuir and first-order kinetics in lower concentrations of formaldehyde, however it should be verified the correlations according to operational parameters. Thermodynamics and isotherm kinetics have been added in the manuscript.

  1. Influence of the

Response: It must be unfinished comments.

  1. Provide and discuss comparative assessment of different AC types and precursor materials to further support understanding and interpretation of the formaldehyde adsorption mechanism

Response: We agree the reviewer’s comment. We also considered the different AC types according to precursors such as coal, biomass, and polymer. Nevertheless, it can be found that the chemical modification of surface is very effective to FA removal. The AC itself is less effective. Therefore, we did not emphasize the effect of AC types according to precursors. Supplementary, we suggested the new synthetic AC for industrial application in the section 5.

  1. A more comprehensive review discussion is needed on formaldehyde indoors adsorption with relevant tables format comparison of different types of AC in relation to influence of the physical and chemical operational parameters on efficiency and maximum adsorption capacity of indoors formaldehyde uptake

Response: As reviewer’s comment, we discussed additionally the physical and chemical operational parameters in the section 2.

Reviewer 3 Report

This review by Kang et al. describes the recent advances in Formaldehyde Removal through Activated Carbon Adsorption The paper addresses a significant overview and organized well. Before publication, I strongly recommend to the authors to add additional section for potential weaknesses and future improvements/perspective. Also, please improve the synthetic scheme representation quality which looks very unprofessional and Please improve figure quality with high resolution.

Author Response

Reviewer 3.

This review by Kang et al. describes the recent advances in Formaldehyde Removal through Activated Carbon Adsorption The paper addresses a significant overview and organized well. Before publication, I strongly recommend to the authors to add additional section for potential weaknesses and future improvements/perspective. Also, please improve the synthetic scheme representation quality which looks very unprofessional and Please improve figure quality with high resolution.

Response: As reviewer’s comment, we added the more discussion in the section 2 with the aspects of kinetics and thermodynamics. In additionally, we totally revised the figures.

Round 2

Reviewer 1 Report

In my opinion, the manuscript still does not meet the review article standards. Very little has been changed in the article. A theoretical part has been added, but it does not correspond to the rest of the text. The reviewers' comments were not addressed seriously. Most of the responses do not address the issues raised. The manuscript still has a number of disadvantages that I pointed out in the first review:

  • The literature review is too limited.
  • The purpose for which the review was carried out was not specified
  • The drawings are of poor quality.
  • In general, no descriptions, explanations and discussion of Figures or Tables were provided.
  • There are no final conclusions.

Author Response

Reviewer 1.

Comment:

In my opinion, the manuscript still does not meet the review article standards. Very little has been changed in the article. A theoretical part has been added, but it does not correspond to the rest of the text. The reviewers' comments were not addressed seriously. Most of the responses do not address the issues raised. The manuscript still has a number of disadvantages that I pointed out in the first review:

  1. The literature review is too limited.

Response: As highlighted in the title, this is “A Brief Review”, but not a review or an overall review. Our brief review focuses on:

1]. Thermodynamics and Kinetics for Adsorption of Formaldehyde on AC

2]. Physical (Operational) Influence Factors

3]. Approaches to Chemical Modifications of AC Surface

4]. Challenge to Industrial Application

  The new version of review manuscript highlights the importance of chemical modifications of AC surface modification by amination treatment and includes 73 references. We cannot add all detail references in this brief review paper. We have made the second revision and highlighted the main ideas in this version. If fact, we cannot buy the reviewer’s comment that “The literature review is too limited”. If we might skip some important references regarding this topic, then, if you suggest literatures to be added in this article, we are willing to look over them and add them in our paper. However, we are very confused about your comment, “too limited”. We would like to ask you, how many literatures will be appropriate for our paper, for instance, more than 100 or more you commented?

  1. The purpose for which the review was carried out was not specified

Response: The article was represented by keywords of activated carbon, formaldehyde removal, surface modification, functional groups, and amination. The purpose of this work was written by authors’ best knowledge regarding the formaldehyde adsorption on the surface of activated carbon. We might mislead the theories or interprets from previous research papers. If this is true, you could have pointed out which parts or paragraphs were misled for future audience. We will be glad if you provide your idea about this comment that the purpose of this work is not clearly presented. We did not claim that this review paper will heavily contribute for this topic but wish to influence the audience whoever is interested in this topic.

Nevertheless, the purpose and focuses of the present review have been revised and highlighted in the manuscript.

  1. The drawings are of poor quality.

Response: We have changed Figure 1 though Figure 4 for better presentation. Additionally, we would like to ask the reviewer to give us her/his aspects about poor quality in detail, so that we can improve them.

  1. In general, no descriptions, explanations and discussion of Figures or Tables were provided.

Response: We referred all Figures and Tables in the manuscript. In general, Figure and Table can deliver the ideas clearer than the written paragraphs. For better presentation, we have revised Figures this time.

  1. There are no final conclusions.

Response: This review article focuses on the formaldehyde adsorption on activated carbon. We understand that activated carbons have been used so diverse areas, such as from drinking water purification to advanced hydrogen storage and CO2 capture. If we cover all the topics regarding the applications of activated carbon, then the contents would be more than dozens of chapters. We also understand that formaldehyde adsorption using activated carbons have been explored since last few decades as well. This paper was intended to summarize the technologies about the surface modifications as well as alternative fabrication of activated carbon, so called bead-type in this article. We believe that the bead-type not granular activated carbon here could be valuable to challenge to commercialization. We agree that the critical details about the bead-type activated carbon were not fully demonstrated but we suggest the capabilities in this article. In summary section, we have added more contents.

Reviewer 2 Report

The authors have significantly improved the quality of the manuscript, yet minor revision is needed for addressing the following

Table 2 and 3: column 5 (description) is repetition of column 2 should be removed and replaced with parameters pH and temperature that infuence sorption mechanism. 

Table 2 and 3: Provide accurate type of model used either linear or non-linear model and if linear what type linear representation used (especially for Langmiur model) 

Section 6: summary part lack significant information and should be more comprehensive in capturing the presented in the work

Author Response

Reviewer 2.

Comment:

The authors have significantly improved the quality of the manuscript, yet minor revision is needed for addressing the following

  1. Table 2 and 3: column 5 (description) is repetition of column 2 should be removed and replaced with parameters pH and temperature that influence sorption mechanism. 

Response: We have revised Table 2 and 3 based on reviewer’s comment.

  1. Table 2 and 3: Provide accurate type of model used either linear or non-linear model and if linear what type linear representation used (especially for Langmuir model) 

Response: Table 2 and Table 3 were presented by thermodynamic isotherm equations and adsorption kinetics, respectively. In the part of isotherm, adsorption amounts were presented according to adsorbate concentration. In Table 2, we added the information of linear and nonlinear patters.

  1. Section 6: summary part lack significant information and should be more comprehensive in capturing the presented in the work

Response: We have added more contents in summary section.

Reviewer 3 Report

Authors revised the manuscript accordingly and can be accepted in the present form.

Author Response

Reviewer 3.

  1. Authors revised the manuscript accordingly and can be accepted in the present form.

Response: We appreciate your recommendation for the publication.

Round 3

Reviewer 1 Report

  1. The literature review is very limited.

Comment: Indeed, the number of 100 references is often taken as a minimum in review manuscripts. This applies to full reviews. I understand that the Authors have accepted a smaller number of cited publications as the manuscript is in the form of a brief-review.

  1. The purpose for which the review was carried out was not specified

Authors’ Response: “The article was represented by keywords of activated carbon, formaldehyde removal, surface modification, functional groups, and amination. The purpose of this work was written by authors’ best knowledge regarding the formaldehyde adsorption on the surface of activated carbon. We might mislead the theories or interprets from previous research papers. If this is true, you could have pointed out which parts or paragraphs were misled for future audience. We will be glad if you provide your idea about this comment that the purpose of this work is not clearly presented. We did not claim that this review paper will heavily contribute for this topic but wish to influence the audience whoever is interested in this topic. Nevertheless, the purpose and focuses of the present review have been revised and highlighted in the manuscript.”

Comment: I asked for an introduction the purposes of review to the manuscript. I will try to briefly explain why. Every research paper should have a research goal (or goals). In the case of review articles, this may be a state-of-art summary, but should be associated with a critical assessment. In the case of technologies, it is valuable to refer to the directions of development of these technologies in the future. A review can be innovative if there is a specific and original aspect to the issue highlighted, especially when this aspect has not been discussed so far. The final conclusions at the end of the manuscript should correspond with the purposes of the manuscript, as stated in the Introduction. Then the text is logically consistent. Therefore, I would ask you to mention the aims of your work clearly at the end of Introduction chapter and to refer to them in the final chapter. For a better understanding, it would also be good to explain the main aspects (chapters) covered in the review. For a better understanding, it would also be good to explain the main aspects (chapters) covered in the review. Here you can explain why was introduced Chapter 2: Thermodynamics and Kinetics for Adsorption of Formaldehyde on AC. This chapter is disconnected from the rest of the text.

  1. The drawings are of poor quality.

Comment: The quality of the drawings has been improved and is now acceptable.

  1. In general, no descriptions, explanations and discussion of Figures or Tables were provided.

Comment: Tables 2 and 3 are still not referenced in the text.

  1. There are no final conclusions.

Comment: Conclusions resulting from the review should correspond to the assumed objectives at the beginning of the text (Introduction section).

Additional comment

  1. Lines 190-192: According to kinetic equations (Table 1), sorption capacity (q), adsorption rate constant (k) and correlation coefficient (R2) were obtained by fitting experimental data.

Comments: Which table do the Authors refer to. There is no table with sorption capacities, adsorption rate constants and correlation coefficients in the text.

Author Response

Reviewer 1.

  1. The literature review is very limited.

Comment: Indeed, the number of 100 references is often taken as a minimum in review manuscripts. This applies to full reviews. I understand that the Authors have accepted a smaller number of cited publications as the manuscript is in the form of a brief-review.

Answer: We highly appreciate the reviewer’s understanding this time.

  1. The purpose for which the review was carried out was not specified

Authors’ Response: “The article was represented by keywords of activated carbon, formaldehyde removal, surface modification, functional groups, and amination. The purpose of this work was written by authors’ best knowledge regarding the formaldehyde adsorption on the surface of activated carbon. We might mislead the theories or interprets from previous research papers. If this is true, you could have pointed out which parts or paragraphs were misled for future audience. We will be glad if you provide your idea about this comment that the purpose of this work is not clearly presented. We did not claim that this review paper will heavily contribute for this topic but wish to influence the audience whoever is interested in this topic. Nevertheless, the purpose and focuses of the present review have been revised and highlighted in the manuscript.”

Comment: I asked for an introduction the purposes of review to the manuscript. I will try to briefly explain why. Every research paper should have a research goal (or goals). In the case of review articles, this may be a state-of-art summary, but should be associated with a critical assessment. In the case of technologies, it is valuable to refer to the directions of development of these technologies in the future. A review can be innovative if there is a specific and original aspect to the issue highlighted, especially when this aspect has not been discussed so far. The final conclusions at the end of the manuscript should correspond with the purposes of the manuscript, as stated in the Introduction. Then the text is logically consistent. Therefore, I would ask you to mention the aims of your work clearly at the end of Introduction chapter and to refer to them in the final chapter. For a better understanding, it would also be good to explain the main aspects (chapters) covered in the review. For a better understanding, it would also be good to explain the main aspects (chapters) covered in the review. Here you can explain why was introduced Chapter 2: Thermodynamics and Kinetics for Adsorption of Formaldehyde on AC. This chapter is disconnected from the rest of the text.

Answer: Thanks for your valuable comments. This comment is good for us to improve our writing of a review paper. We have further modified the introduction section and summary section. Some modified paragraphs in summary have been added to the introduction. And last paragraph in introduction has been revised to introduce each sections from section 3 through 5.

In section 2, we summarized thermodynamics and kinetics as pointed out above. In this section, we have to add them because the reviewer 2 strongly suggested to add thermodynamics and kinetics of formaldehyde adsorption behaviors when we prepare 2nd revision. It looks little deviated from the main focus of our brief review, however, it also provides the information about formaldehyde on activated carbons. As referred in other thermodynamics and kinetics studies, formaldehyde adsorptions are not exceptionally different. This section 2 also presents the similar approaches to these topics.

  1. The drawings are of poor quality.

Comment: The quality of the drawings has been improved and is now acceptable.

Answer: We highly appreciate the reviewer’s understanding.

  1. In general, no descriptions, explanations and discussion of Figures or Tables were provided.

Comment: Tables 2 and 3 are still not referenced in the text.

Answer: We have revised to correct them.

  1. There are no final conclusions.

Comment: Conclusions resulting from the review should correspond to the assumed objectives at the beginning of the text (Introduction section).

Answer: Please refer to the previous responses above.

  1. Lines 190-192: According to kinetic equations (Table 1), sorption capacity (q), adsorption rate constant (k) and correlation coefficient (R2) were obtained by fitting experimental data.

Comments: Which table do the Authors refer to. There is no table with sorption capacities, adsorption rate constants and correlation coefficients in the text.

Answer: Line2 190-192 were the typos included. We have changed them.
